# Influence of Surgical Procedures on C-Reactive Protein Levels in Severely Burned Patients: Preliminary Analysis on Implications for Early Sepsis Diagnosis

**DOI:** 10.3390/ijms26115158

**Published:** 2025-05-28

**Authors:** Małgorzata Barbara Makowska-Rezaie, Michał Jeleń, Marzenna Bartoszewicz, Tomasz Korzeniowski, Maria Kamila Klimeczek-Chrapusta, Anna Marta Chrapusta

**Affiliations:** 1Malopolska Burn and Plastic Surgery Center, Ludwik Rydygier Specialized Hospital, 31-826 Krakow, Poland; 2Department of Immunopathology, Molecular Biology Faculty of Pharmacy, Wroclaw Medical University, 50-368 Wroclaw, Poland; 3Department of Pharmaceutical Microbiology and Parasitology, Faculty of Pharmacy, Wroclaw Medical University, 50-368 Wroclaw, Poland; 4East Center of Burns Treatment and Reconstructive Surgery, 21-010 Łęczna, Poland; 5Department of Plastic, Reconstructive Surgery and Burn Treatment Medical University of Lublin, 20-059 Lublin, Poland; 6Faculty of Medicine, Medical College, Jagiellonian University, 30-688 Kraków, Poland; 7Department of Pediatric Surgery, University Children’s Hospital of Krakow, 30-663 Kraków, Poland

**Keywords:** sepsis, burns, inflammation, necrectomy, wound healing

## Abstract

The local treatment of deep burn wounds involves the excision of the necrosis and covering the wounds with skin grafts. Surgical procedures are thought to have an impact on the inflammatory response, especially in severe burn patients requiring treatment in an intensive care unit. Currently, there are no available data in the literature regarding the correlation of the type of surgical procedure and the levels of the inflammatory markers. This study investigates the importance of monitoring c-reactive protein (CRP) around the time of surgical burn procedures and how it can aid in assessing the inflammatory response. Of the 810 burn patients, 93 patients aged 20 to 74 years with IIb- and III-degree burns covering 20% to 50% of the total burned body surface were included in this prospective study. Three subgroups were recognized based on the surgical procedure performed: fascial necrectomy, tangential necrectomy, and skin grafting. The research material included blood samples collected in the early postoperative hours. A total of 270 CRP level measurements were performed. A reduction in CRP levels was observed three hours post-procedure in patients who underwent skin harvesting for grafting. Conversely, a significant increase in CRP levels was noted between postoperative timepoints in patients who underwent tangential necrectomy.

## 1. Introduction

Large surface area, full-thickness skin burns represent one of the most severe forms of injury. They elicit a profound stress response to trauma and serve as a potent trigger for initiating the inflammatory cascade.

Aside from the burn injury itself, surgical procedures also can lead to the disruption of the body’s homeostasis [1,2]. In cases of deep third-degree burns caused by fire, the excision of necrotic tissue down to the fascial layer is performed, typically involving the removal of burned skin and subcutaneous fat using electrocoagulation. For more superficial burns, a tangential excision is carried out with a dermatome, which removes necrotic tissue layer by layer until a viable, bleeding surface is reached. Skin graft harvesting, also performed with a dermatome, enables the closure of burn wounds and promotes the healing of the affected area [1,2,3,4].

In burn patients who undergo multiple surgical procedures, an inflammatory response can be observed in the perioperative period [2]. This response to iatrogenic trauma can have profound consequences for wound healing and the susceptibility to infection and cause disruptions in organ functions.

Bacterial infections in burn wounds have been recognized as a clinical challenge for over a century. Techniques such as the early excision of necrotic tissue and rapid wound coverage with skin grafts are effective in reducing those risks. When it comes to severe burns where an immediate necrosis excision and wound closure are not feasible, the close monitoring of inflammatory markers can be essential for future recovery, considering the heightened risk of sepsis [1,3,4].

The detection of infections and the early identification of sepsis are particularly critical in patients with severe burns, which is why the need for a rapid and reliable method to diagnose bloodstream infections has been a subject of ongoing discussion. Although guidelines for the diagnosis and recognition of sepsis have undergone several revisions, they are primarily based on data from the general population. Burn patients, however, are different, due to their unique hyper-metabolic response and the associated clinical and laboratory abnormalities. Consequently, this population was excluded from the foundational studies that informed the development of the SEPSIS-3 definition and the Surviving Sepsis Campaign guidelines [5,6].

Sepsis in burn patients exhibits distinct pathophysiological characteristics, rendering standard diagnostic criteria often ineffective in clinical practice, despite their widespread recommendation [7]. This presents a major challenge in the management of severe burns, where infection is the most common complication and sepsis remains the leading cause of mortality, accounting for 50–84% of all adult deaths in burn centers [8]. In addition to clinical evaluation, laboratory blood markers serve as essential tools for assessing the risk of systemic infection and sepsis. Among these, CRP, a widely available positive acute-phase reactant, is commonly used to monitor the inflammatory response and remains a valuable component of the diagnostic process [4].

CRP, discovered in 1930 by Tillet and Francis, turned out to be a marker of the tissue damage and inflammation of both infectious and non-infectious origins [1,9]. Produced mainly in hepatocytes, it is stimulated by other inflammatory cytokines, primarily IL-6. Moreover, it is also produced by the endothelial cells, adipocytes, and macrophages of the atherosclerotic plaque [2]. It exhibits both pro- and anti-inflammatory effects. It promotes the clearance of pathogens and damaged cells through complement activation and phagocytosis, but excessive activation can contribute to tissue injury and cytokine release. CRP also has prothrombotic properties and binds to LDL particles, facilitating their uptake by macrophages [4]. Its vascular effects include the inhibition of eNOS, the upregulation of angiotensin II receptors, and increased endothelin levels, impairing vasodilation [5]. The use of CRP measurements in the prognosis of sepsis has been widely described in the available literature [10,11,12,13].

While aggressive surgical treatment is known to influence the inflammatory response in burn patients, the impact of specific procedures—such as necrosis excision and skin grafting—on inflammatory markers like CRP remains unclear. This study hypothesizes that surgical interventions, including fascial excision, tangential excision, and split-thickness skin grafting, may alter CRP levels. Rather than focusing on absolute CRP values, we aimed to examine their fluctuations, as these changes could affect the reliability of CRP as an early postoperative predictor of sepsis. To our knowledge, this is the first study to explore this relationship in the context of burn care.

## 2. Results

### 2.1. Study Group Characteristics

Out of 810 patients, 93 met the inclusion and exclusion criteria and were selected for this study; however, 3 were excluded due to missing laboratory values. The study population consisted of 88% men and 12% women, aged 20 to 74 years. The mean age was 47.6 years, with a median of 50 years. The average burn area was 33.1% of the TBSA and the median was 35%. The hospitalization ranged from 19 to 292 days, with a mean length of stay of 79.4 days. Inhalation injuries were identified in 10 patients, and 6 had multiple chronic comorbidities. Clinical guidelines for the choice of surgical procedure were followed [1,4,9]. The study group characteristics are in Table 1.

A total of 270 CRP measurements were collected from the study groups at 3 and 9 h after surgery. The control group consisted of the same 90 patients on non-operative days, who met identical inclusion and exclusion criteria, with CRP levels measured on days when no surgical intervention was performed.

### 2.2. CRP Levels in Fascial Necrectomy Group

A total of 87 CRP measurements were collected from 29 patients undergoing fascial necrectomy, with each patient having CRP levels measured at three timepoints, 3 and 9 h after surgery, compared to CRP levels obtained from the same individuals during non-operative periods, serving as controls. The distribution of the CRP values at all timepoints met the assumptions of normality based on the Shapiro–Wilk test (*p* > 0.05), which was visually confirmed by Q-Q plots. The repeated measures analysis using the Friedman test showed no statistically significant changes in CRP levels across the three timepoints (*p* = 0.905). Post hoc comparisons using Wilcoxon signed-rank tests with a Bonferroni correction (α = 0.0167) also revealed no significant pairwise differences. The results for this group are presented in Table 2. The normality of the data distribution is presented with a Q-Q plot in Figure 1. The Box plot showing CRP levels measured at three timepoints—during a non-operative control period and at 3 and 9 h following fascial necrectomy—and the Line plot (spaghetti plot) illustrating individual patient CRP trajectories over time are presented in Figure 2.

### 2.3. CRP Levels After Tangential Necrectomy

A total of 87 CRP measurements, taken at 3 and 9 h after tangential necrectomy, were analyzed in a study group of 29 patients and compared to CRP levels obtained from the same individuals during non-operative periods, serving as controls. The distribution of CRP values did not meet the assumptions required for the ANOVA. Therefore, the Friedman test was applied and revealed significant differences in CRP levels (*p* = 0.0087). The post hoc analysis using paired Wilcoxon signed-rank tests with a Bonferroni correction (α = 0.0167) demonstrated a significant increase in CRP at 9 h compared to the 3 h measurement (*p* = 0.0027), while differences between the control vs. 3 h and the control vs. 9 h were not statistically significant. The observed rise in the CRP concentration at 9 h post-procedure is a notable finding. The results for this group are presented in Table 3. The normality of the data distribution is present with the Q-Q plot in Figure 3. CRP concentrations in the tangential necrectomy group illustrating CRP levels measured during a non-operative control period and at 3 and 9 h after a tangential necrectomy and the plot displaying individual patient CRP trajectories across the three timepoints are presented in Figure 4.

### 2.4. CRP Level After Skin Graft Harvesting

A total of 96 CRP measurements were analyzed in a study group of 32 patients undergoing split-thickness skin graft harvesting. Each patient had CRP values measured at three timepoints: during a non-operative period (serving as the control) and at 3 and 9 h following the procedure. The Shapiro–Wilk test confirmed that the CRP data at all timepoints deviated significantly from a normal distribution (*p* < 0.001), which was also supported by the Q–Q plot inspection. Consequently, the non-parametric Friedman test was used and revealed a statistically significant difference in CRP levels across the three timepoints (*p* = 0.040). The post hoc analysis using Wilcoxon signed-rank tests with a Bonferroni correction (α = 0.0167) showed a statistically significant decrease in CRP levels at 3 h compared to the control (*p* = 0.0165), while no significant differences were found between the control and 9 h values or between the 3 and 9 h values. The CRP results for this group are presented in Table 4. The normality of the data distribution is present with the Q-Q plot in Figure 5. CRP concentrations in the skin graft harvesting group Box plot showing CRP levels measured during a non-operative control period and at 3 and 9 h after skin graft harvesting and the Line plot illustrating individual patient CRP trajectories across the three timepoints are presented in Figure 6.

## 3. Discussion

This study’s results suggest that the type of surgical intervention in burn treatments influences short-term fluctuations in CRP levels. The fascial necrectomy did not significantly alter CRP concentrations postoperatively, suggesting it may have a minimal impact on the systemic inflammatory response. In contrast, the tangential necrectomy was associated with a significant increase in CRP levels, particularly at 9 h post-procedure, indicating a more pronounced inflammatory activation. Split-thickness skin grafting, on the other hand, resulted in a slight but statistically significant decrease in CRP at 3 h, suggesting a transient suppression or stabilization of inflammation.

This study has several limitations. Measuring CRP levels at 3 and 9 h postoperatively does not reflect typical CRP kinetics, which peak around 48 h. However, in the context of severe burn care, standard timing is often impractical due to the need for repeated surgical interventions within short intervals. In clinical practice, necrotic tissue is frequently excised and the wound is closed with a skin graft during the same surgical session, or in staged procedures performed shortly after one another to reduce the infection risk. As a result, assessing CRP levels at 24 or 48 h postoperatively is not feasible in many cases. Therefore, this study focused on the early postoperative period, recognizing that CRP dynamics in burn patients must be interpreted within the specific context and timing of surgical treatments [1,2,3,4,5,6,7,8,9,10,14].

Another limitation is the small sample size, with only 93 patients divided across three study groups. This restricts the statistical power and prevents definitive conclusions. The findings should be viewed as preliminary observations rather than confirmed outcomes.

Despite its limitations, this study suggests that CRP levels vary by surgical type, with tangential necrectomy showing the greatest increase—likely reflecting higher surgical trauma. The fascial necrectomy had no significant effect, indicating differing systemic responses. This study’s strength lies in its novel focus on early CRP dynamics following specific burn surgeries, supporting the need for the procedure-specific interpretation of inflammatory markers [1,2,3,4,5,6,7,8,9,10,11,12,13,14,15,16,17,18,19,20,21,22,23,24].

As previously mentioned, trauma is a known cause of CRP elevation [14]. The increase is gradual, beginning at around 2 h after the triggering event, becoming more noticeable at 24 h, and potentially rising up to 100-fold by 48 h. If no CRP elevation is observed within 24 h of clinical symptoms, infection can likely be ruled out. CRP levels between 10 and 100 mg/L typically indicate localized inflammation, while values between 100 and 1000 mg/L suggest a generalized or severe inflammatory response [19,20,21,22,23,24]. When the damaging stimulus ceases or proper therapy is administered, the CRP concentration decreases rapidly and returns to physiological values within 3–7 days [9,15,16,17].

CRP levels in this study were measured using the immunoturbidimetric method on the DxC 700 AU analyzer (Beckman Coulter, Brea, CA, USA). This analyzer detects protein concentrations based on changes in the light scattering and transmittance caused by antigen–antibody complex formations. The assay duration is approximately 60 min, providing reliable and timely results. Immunoturbidimetric, immunoenzymatic, and laser nephelometric techniques are among the most commonly used and widely available methods for CRP quantification and are valued for their low cost, precision, and efficiency. For detecting lower CRP levels, high-sensitivity assay variants are recommended [18].

Sepsis is a serious epidemiological and therapeutic problem. A large analysis by Gül et al. in 2020 [15] reported 48.9 million cases with 11 million deaths from sepsis in 2017. According to the study, one in five deaths is caused by sepsis, 85% of them occur in developing countries, and two in five occur in children under the age of 5. In the 20th century, the significant mortality caused by the presence of bacteria in the blood prompted many studies on sepsis and the influence of the patient’s immune response on its course. The heterogeneity of the disease process and the lack of a clear definition of sepsis created serious difficulties in its recognition and treatment as well as in interpreting the results of the relevant published research [15].

The current definition of sepsis, known as SEPSIS-3, was introduced in 2016 and published in the *Journal of the American Medical Association*. It defines sepsis as a “life-threatening organ dysfunction caused by a dysregulated host response to infection” [19,20]. An inherent element of the course of severe trauma is SIRS, an inflammatory reaction occurring in response to tissue damage and blood loss, which develops within the first 30 min after the injury [17]. Serious injuries often cause significant immunosuppression, triggering a systemic, acute, and nonspecific immune response, paradoxically resulting in a decreased resistance to infections. The cause is a rapid, massive loss of T lymphocytes by apoptosis. The mechanism of this phenomenon remains unclear [19,20]. Lymphopenia is preceded by the occurrence of inflammation with a very high concentration of proinflammatory cytokines [20]. The ratio of proinflammatory to anti-inflammatory factors determines whether the body will be able to restore homeostasis or, otherwise, enter a state of a chronic inflammatory response with immunosuppression and hypermetabolism [21]. The release of mediators depends primarily on the severity of the injury and secondarily on the activation of different mediator cascades during post-traumatic or postoperative complications. Mediators are therefore of crucial importance with respect to the intensity of organ damage and the outcome of treatments.

Injury leads to the release of alarmins from damaged tissues. Alarmins react with immune cells to initiate inflammation. Inflammatory responses to injury are primarily mediated by macrophages. Cellular elements released due to their damage, together with the stimulated endothelium, cause the activation of the coagulation cascade. Following the injury, the complement system is activated mainly through the elements of the coagulation cascade. The effects of uncontrolled complement activation can lead to systemic tissue and organ damage. A properly functioning inflammatory reaction is self-limiting and self-terminating. As mentioned, trauma-induced inflammation can cause an excessive sensitivity of immune system cells, mainly macrophages and neutrophils. These cells may become overreactive to bacterial antigens and toxins produced by bacteria and, by reacting with them, initiate a massive release of proinflammatory cytokines [18,22]. The anti-inflammatory response, largely regulated by T leukocytes, acts as a control mechanism preventing excessive inflammation [23]. The anti-inflammatory response may actively suppress antimicrobial immunity and thus be responsible for the increased susceptibility to infections of trauma patients [22].

Surgeries necessary for treatment are an additional iatrogenic trauma that may complicate the disease course by inducing secondary inflammatory reactions [25].

Changes in the CRP level are believed to reflect not only the state of inflammation but also the response to surgical trauma [26]. The assessment of CRP in the perioperative period was performed by Iizuka et al. The authors assessed its levels in 80 patients who underwent osteosynthesis due to mandibular fractures. In their study each surgical procedure resulted in a significant increase in CRP values. In 82% of patients, this parameter reached its highest value on the second day post-procedure, and later the values gradually decreased. Similar conclusions were drawn by Neumaier et al. They conducted a prospective study on a group of 1418 patients undergoing the open reposition and stabilization of fractures in the upper or lower extremities. Blood was collected upon admission, before the procedure, and then three times during the first 12 days after the procedure. They noted a statistically significant increase in CRP after the procedure, with the greatest rise on the second day [27]. Another study evaluating CRP levels in the perioperative period was conducted by Cole et al. on a group of 201 patients undergoing elective procedures in a general surgery department. The tests were performed before the procedure and for the following 5 days after the procedure. The study showed an increase in CRP values after the procedure, with a peak on the second day [28].

The existing literature confirms a predictable increase in CRP levels following an iatrogenic trauma, such as surgery; however, there is a lack of data regarding the behavior of this marker in burn patients. The complex nature of burn wounds and their surgical management, which differs significantly from other clinical scenarios, provided the rationale for conducting this study.

This study’s results differ from the conclusions cited above. The authors hypothesize that the observed decrease in CRP levels following skin graft harvesting may be attributed to the closure of the burn wound, which could lead to a reduction in the systemic inflammatory response. However, based on this reasoning, one might expect a similar response following necrotic tissue removal, regardless of the surgical method used—an outcome that was not observed. Notably, both the tangential excision of necrosis and skin graft harvesting are forms of iatrogenic injury, both performed with the dermatome. Given that the dermatome blade traverses the same anatomical layer—the dermis—in both procedures, a comparable inflammatory response might be anticipated. Contrary to this expectation, this study revealed divergent outcomes: CRP levels increased after the tangential necrectomy but decreased following skin graft harvesting.

Although the observed decrease in CRP levels following skin grafting falls within the range of normal biological variability [19,20,21,22,23,24], it appears to be clinically relevant. While the small sample size is a recognized limitation, the authors believe these preliminary findings are noteworthy. A decrease in CRP after grafting may serve as a useful prognostic indicator, whereas an unexpected increase could signal a heightened risk of infection. Recognizing this pattern may aid in the more cautious interpretation of CRP fluctuations throughout the treatment course of burn patients. The authors suspect that, for instance, a rise in CRP following a tangential necrectomy may mimic early sepsis, potentially prompting an unnecessary pharmacological intervention. Conversely, failing to recognize the typical decline in CRP after skin grafting could lead to missed signs of bacterial translocation and inflammation.

These initial findings have encouraged the authors to continue this line of research, incorporating additional parameters and expanding the patient cohort. Further studies may clarify the role of CRP in predicting inflammation or sepsis in the perioperative management of burn patients and support the development of more precise clinical guidelines.

## 4. Materials and Methods

### 4.1. Study Population

Study population consisted of patients hospitalized in the Burn Unit of the Małopolska Plastic Surgery and Burn Center in Kraków between 2019 and 2023, selected based on following inclusion and exclusion criteria.

### 4.2. Inclusion and Exclusion Criteria

Inclusion criteria included the following:Ages between 18 and 75;Thermal injury;Second b- and third-degree burns, with an area of no less than 20% and no more than 50% of total body surface area (TBSA);Patients who underwent only one of the following surgical procedures during a single operation: (1) tangential necrectomy, (2) fascial necrectomy, or (3) split-thickness skin graft harvesting;The surgical procedure encompassed 5–20% of TBSA.

Exclusion criteria included the following:Electric or chemical injury;Patients who underwent more than one of the listed above surgical procedures during a single operation;Surgical procedure performed on an area of less than 5% and greater than 20% TBSA;Diagnosis of a concomitant disease that significantly affected the immune response;Immune deficiency;Immunosuppressive treatment at the time of injury.

### 4.3. Study Design

Patients were divided into three intervention groups and one control group based on the type of surgical procedure performed. Each intervention group consisted of CRP level measurements taken after a specific intervention:Group I: tangential necrectomyGroup II: fascial necrectomyGroup III: split-thickness skin harvesting for grafting

In all study groups, venous blood samples were collected at 3 and 9 h postoperatively to measure CRP levels. Tangential excision for deep second-degree burns and split-thickness skin harvesting were performed using a dermatome, while fascial excision for third-degree burns was carried out using electrocoagulation [19,20].

The control group consisted of the same patients, all of whom met identical inclusion and exclusion criteria, with matching age range and burn extent. Control measurements were taken on non-operative days—specifically when no surgical procedure had been performed within the preceding 48 h—with CRP levels assessed under these conditions.

### 4.4. Material Collection and CRP Analysis

Prior to blood collection, each patient provided written informed consent, which outlined the study’s purpose, procedures, potential risks, and benefits. Blood samples were drawn from an unburned area using a separate peripheral intravenous line or a central venous catheter. Standard preparation included positioning the patient comfortably and safely, followed by antiseptic cleansing of the puncture site with an alcohol-based solution. In healthy individuals, CRP levels are typically <5 mg/L, a concentration often referred to as being at or below the limit of detection.

A sterile, closed vacuum system (BD Vacutainer) was used for venous sampling. Tubes were labeled with the patient’s information before collection. A total of 4 mL of blood was obtained and securely transported to the laboratory. CRP levels were measured using the immunoturbidimetric method on the DxC 700 AU analyzer (Beckman & Coulter, Singapore). The test, which takes approximately 60 min, detects protein concentration based on changes in light scattering and transmittance.

### 4.5. Ethical Statement

The Declaration of Helsinki’s principles were respected. This study received approval by the Ethic Committee of the Medical University in Wroclaw, approval number: KB 133/2019, Approval Date: 28 February 2019.

### 4.6. Statistical Analysis

The results were analyzed using appropriate statistical methods. Normality of data distribution was assessed using the Shapiro–Wilk test and visually confirmed with quantile–quantile (Q–Q) plots. Temporal changes in CRP levels were evaluated using repeated measures ANOVA when assumptions of normality and sphericity were satisfied; otherwise, the non-parametric Friedman test was applied. Post hoc analyses were performed using paired Student’s *t*-tests with Bonferroni correction following ANOVA and paired Wilcoxon signed-rank tests with Bonferroni correction following the Friedman test to adjust for multiple comparisons.

Comparisons among surgical procedure groups were conducted using one-way ANOVA when parametric assumptions were met and the Kruskal–Wallis test otherwise.

## 5. Conclusions

The interpretation of CRP values in burn patients requires caution due to the influence of multiple overlapping factors, including the inflammatory response to surgical interventions. This study revealed that tangential necrectomy is associated with a significant increase in CRP levels, particularly at 9 h post-procedure, suggesting a stronger systemic inflammatory response. In contrast, the fascial necrectomy did not result in notable CRP fluctuations, while split-thickness skin grafting was followed by a modest but statistically significant decrease in CRP at 3 h.

Although this study is limited by its small and heterogeneous sample size, which prevents definitive statistical conclusions, the observed trends are novel and may provide a foundation for further research.

## Figures and Tables

**Figure 1 ijms-26-05158-f001:**
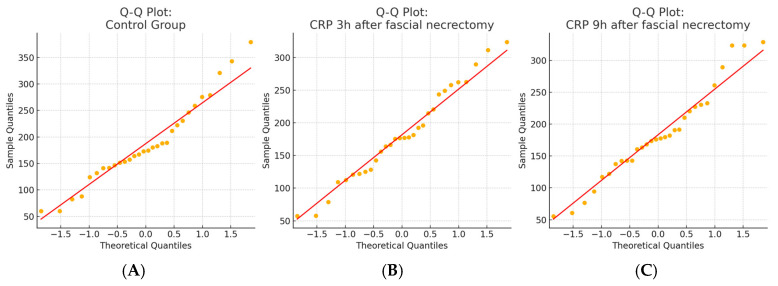
Q–Q plots of CRP distributions in the fascial necrectomy group. (**A**) Control group, (**B**) 3 h after fascial necrectomy, and (**C**) 9 h after fascial necrectomy. The Shapiro–Wilk test confirmed the normal distribution for all timepoints (*p* > 0.05), which was also supported by a visual assessment using Q–Q plots.

**Figure 2 ijms-26-05158-f002:**
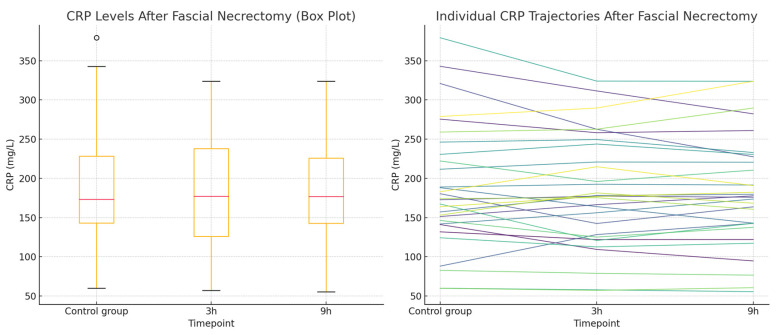
CRP concentrations in the fascial necrectomy group. (**Left**): Box plot showing CRP levels measured at three timepoints—during a non-operative control period and at 3 and 9 h following fascial necrectomy. The plot displays the median, interquartile range, and full range of values. (**Right**): Line plot (spaghetti plot) illustrating individual patient CRP trajectories over time.

**Figure 3 ijms-26-05158-f003:**
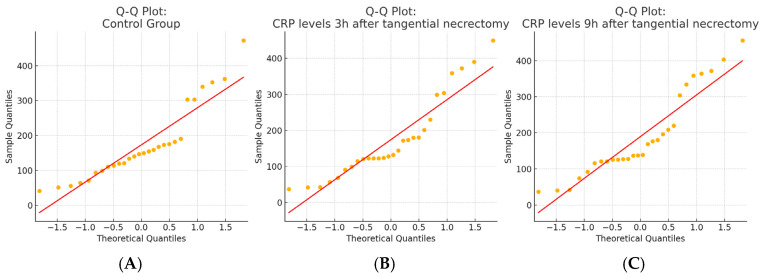
Q–Q plots of CRP level distributions in each study group. (**A**) The control group; (**B**) 3 h after tangential necrectomy; and (**C**) 9 h after tangential necrectomy. Normality was assessed visually using Q–Q plots and statistically using the Shapiro–Wilk test, which indicated non-normal distributions in all three groups (*p* < 0.05).

**Figure 4 ijms-26-05158-f004:**
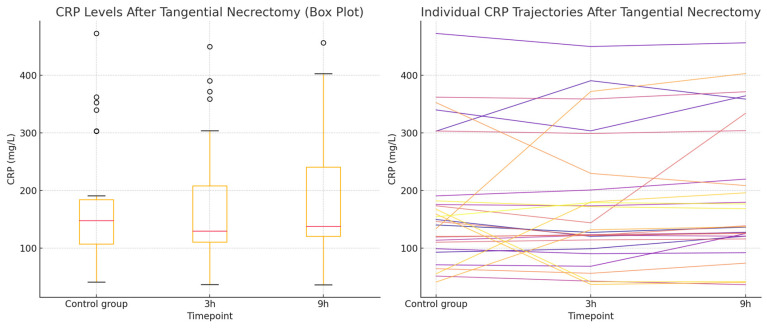
CRP concentrations in the tangential necrectomy group. (**Left**): the Box plot illustrating CRP levels measured during a non-operative control period and at 3 and 9 h after the tangential necrectomy. The plot shows the median, interquartile range, and full data range. (**Right**): the Line plot displaying individual patient CRP trajectories across the three timepoints.

**Figure 5 ijms-26-05158-f005:**
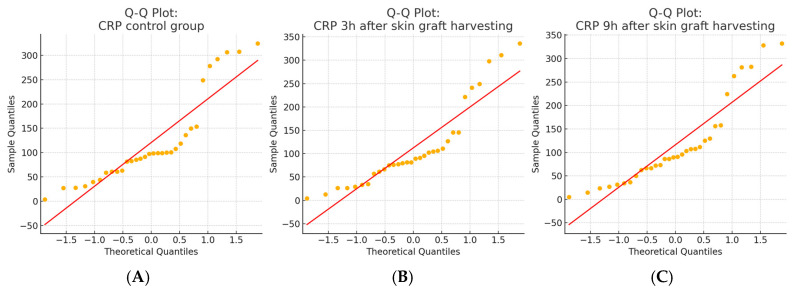
Q–Q plots of CRP values in the skin graft harvesting group. (**A**) the CRP control group; (**B**) CRP levels 3 h after skin graft harvesting; and (**C**) CRP levels 9 h after skin graft harvesting. The Shapiro–Wilk test indicated that CRP values in all three groups deviated significantly from the normal distribution (*p* < 0.001), which was also confirmed by a visual inspection of the Q–Q plots. These results justified the use of non-parametric statistical methods in subsequent analyses.

**Figure 6 ijms-26-05158-f006:**
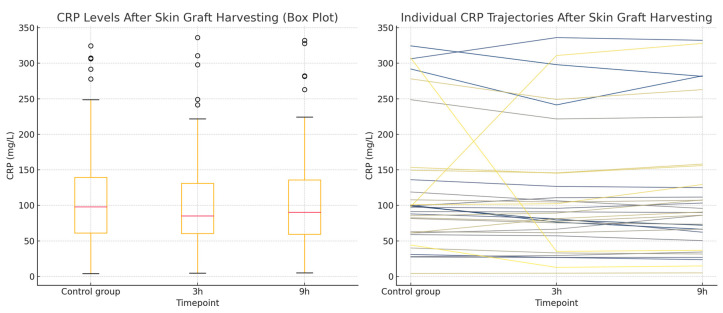
CRP concentrations in the skin graft harvesting group. (**Left**): a Box plot showing CRP levels measured during a non-operative control period and at 3 and 9 h after skin graft harvesting. The plot displays the median, interquartile range, and total range of values. (**Right**): a Line plot illustrating individual patient CRP trajectories across the three timepoints.

**Table 1 ijms-26-05158-t001:** Study group characteristics.

Feature	*N*
Total patients included	93
Age (years), mean ± SD	47.6 ± 15.0
Age, median (range)	50 (20–74)
Gender distribution	88% male, 12% female
TBSA (%), mean ± SD	33.1 ± 8.9
TBSA, median	35%
% II-degree burns, mean ± SD	4.9 ± 6.6
% III-degree burns, mean ± SD	22.6 ± 7.0
Inhalation injuries	10 patients
Chronic comorbidities	6 patients
Length of hospitalization (days), mean ± SD	79.4 ± 52.1
Length of hospitalization, range	19–292

**Table 2 ijms-26-05158-t002:** CRP levels in the group of the necrectomy to the fascia.

	Time	Control Group	Study Group 3 h Post Op	Study Group 9 h Post Op
Parameter				
Minimum		59.57	57.17	55.48
Maximum		379.32	324.08	329.10
Average		187.42	181.97	183.59
Median		173.47	177.10	176.85
Standard deviation		77.39	71.21	72.99
Comparison analysis				
Comparison		*p*-value	Significance	
Control vs. 3 h		0.44	Not significant	
Control vs. 9 h		0.59	Not significant	
3 h vs. 9 h		0.53	Not significant	

**Table 3 ijms-26-05158-t003:** CRP levels in tangential necrectomy group.

	Time	Control Group	Study Group 3 h Post Op	Study Group 9 h Post Op
Parameter				
Minimum		41.21	36.97	36.63
Maximum		427.88	450.16	456.62
Average		173.16	173.54	188.23
Median		148.06	131.87	138.90
Standard deviation		108.49	111.23	115.86
Comparison analysis				
Comparison		*p*-value	Significance	
Control vs. 3 h		0.35	Not significant	
Control vs. 9 h		0.19	Not significant	
3 h vs. 9 h		0.0027	Significant	

**Table 4 ijms-26-05158-t004:** CRP level in skin grafting group.

	Time	Control Group	Study Group 3 h Post Op	Study Group 9 h Post Op
Parameter				
Minimum		4.13	4.54	4.98
Maximum		324.50	336.16	332.32
Average		115.62	110.20	113.91
Median		94.33	85.26	90.24
Standard deviation		91.01	87.00	89.2992
Comparison analysis				
Comparison		*p*-value	Significance	
Control vs. 3 h		0.0165	Significant	
Control vs. 9 h		0.4	Not significant	
3 h vs. 9 h		0.12	Not Significant	

## Data Availability

The data presented in this study are available on request from the corresponding author.

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
