# Peer review of "Influence of Surgical Procedures on C-Reactive Protein Levels in Severely Burned Patients: Preliminary Analysis on Implications for Early Sepsis Diagnosis"

_ijms, 2025, doi:10.3390/ijms26115158_

Round 1
Reviewer 1 Report
Comments and Suggestions for Authors
This study evaluates how different surgical treatments for deep burns influence postoperative inflammation by monitoring CRP levels, helping assess inflammatory responses in ICU patients with severe burn injuries. Although the paper presents a certain scientific interest, there are some concerns regarding the validity and the overall results. Here are some important comments:
- The title is weak and too long; please change the paper title to fit the content. You can use this: “Influence of Surgical Procedures on C-Reactive Protein Levels in Severely Burned Patients: Implications for Early Sepsis Diagnosis – A Preliminary Analysis”.
- Lines 88–89 of the hypothesis are unclear and inconsistent with the findings. Although the results clearly show procedure-dependent changes, the authors argue that surgery shouldn't have an impact on CRP. Rephrase the hypothesis to consider the study's exploratory nature.
- The 34-person sample size is insufficient to allow for the drawing of generalizable conclusions. Incorporate power analysis into the discussion, or at the very least, recognize it as a more significant constraint.
- It is unclear whether the 93 CRP measurements in the control group are from 93 different patients or a combination of measurements from fewer patients. Make this clear and match the study group with patient demographics (such as age and TBSA).
- When performing repeated post-hoc Wilcoxon tests, the manuscript does not employ a multiple comparison correction strategy. Make the necessary corrections and report them (e.g., Bonferroni).
- Measuring CRP three and nine hours after surgery is inconsistent with established CRP kinetics, which peak at around 48 hours. Describe this methodology or add more timepoints in future studies.
- The authors consider a drop in CRP after grafting to be clinically significant, even though this falls within normal biological variation. Provide effect sizes and reference ranges to support clinical relevance.
- Although sepsis is frequently discussed, none of the patients in this study were diagnosed with it. Avoid overinterpreting the data or separating exploratory results from possible sepsis implications in this case.
- Minor typographical errors must be revised; the English language needs more improvement.
Author Response
Response to Reviewer 1 Comments -
First of all, I would like to thank you very much for taking the time and effort to review this manuscript. Please find the detailed responses below and the corresponding corrections highlighted in red fonds in the re-submitted files.
All changes in the text consistent with the comments of reviewer 1 are in red font.
- The title is weak and too long; please change the paper title to fit the content. You can use this: “Influence of Surgical Procedures on C-Reactive Protein Levels in Severely Burned Patients: Implications for Early Sepsis Diagnosis – A Preliminary Analysis”.
I changed the title.
- Lines 88–89 of the hypothesis are unclear and inconsistent with the findings. Although the results clearly show procedure-dependent changes, the authors argue that surgery shouldn't have an impact on CRP. Rephrase the hypothesis to consider the study's exploratory nature.
The research hypothesis was formulated differently (lines 87-89).
- The 34-person sample size is insufficient to allow for the drawing of generalizable conclusions. Incorporate power analysis into the discussion, or at the very least, recognize it as a more significant constraint.
At the beginning of the discussion, the weak point of the small number of patients studied was additionally highlighted. (218-220)
I found my error – insted of „34 patients” should be „93 patients”. I changed it in the latest version of text.
- It is unclear whether the 93 CRP measurements in the control group are from 93 different patients or a combination of measurements from fewer patients. Make this clear and match the study group with patient demographics (such as age and TBSA).
The control group is briefly described in lines 371-375.
- When performing repeated post-hoc Wilcoxon tests, the manuscript does not employ a multiple comparison correction strategy. Make the necessary corrections and report them (e.g., Bonferroni).
The statistical analysis was improved according to the reviewer's suggestions.
- Measuring CRP three and nine hours after surgery is inconsistent with established CRP kinetics, which peak at around 48 hours. Describe this methodology or add more timepoints in future studies.
We know that CRP measurement three and nine hours after surgery is inconsistent with the established CRP kinetics, which peaks at about 48 hours. However, in the treatment of severe burns, measurement 48 hours after the primary surgery is not possible in this clinical study because then or 24 hours earlier another surgical procedure is performed. Therefore, standard testing procedures cannot be applied in this group of patients and the kinetics of the CRP marker should be studied according to the specifics of surgical treatment of burns.
The explanation is given in lines 208-217.
- The authors consider a drop in CRP after grafting to be clinically significant, even though this falls within normal biological variation. Provide effect sizes and reference ranges to support clinical relevance.
The significance of the information on the decreasing trend of CRP after skin graft surgery in clinical practice is further explained in lines 322-325.
- Although sepsis is frequently discussed, none of the patients in this study were diagnosed with it. Avoid overinterpreting the data or separating exploratory results from possible sepsis implications in this case.
I kindly ask for your consent to leave the implications related to sepsis because in the treatment of severe burns, the problem of sepsis is key to the systemic treatment and monitoring of the patient.
- Minor typographical errors must be revised; the English language needs more improvement.
Language errors have been corrected.
Reviewer 2 Report
Comments and Suggestions for Authors
This is an interesting study that explores CRP levels following different surgical procedures for burns. It is an important and clinically relevant area of research.
Introduction
In the introduction I suggest the authors add more details on the procedure of excision of necrosis to the fascia, tangential excision of necrosis, and split thickness skin grafting so that the reader can appreciate more what these procedures involve.
In the introduction, I suggest the authors make the connection between CRP levels and sepsis clearer. For example, include, if possible, more references that provide evidence that CRP levels can be used to predict sepsis.
Materials and Methods
The materials and methods do not state how many patients were in each study group, when the blood samples were collected and from which body site.
The methods section could contain further information about informed consent.
Suggest expanding the study to include more participants. Cytokine levels or other parameters could also be measured.
Results
Suggest presenting the results as charts showing mean CRP levels +/- standard deviation.
Numbers should be written with a decimal point rather than a comma.
The values that are statistically significant could be more clearly marked in the results section.
The conclusions seem to be supported by the results and references are relevant.
Author Response
Response to Reviewer 2 Comments -
First of all, I would like to thank you very much for taking the time and effort to review this manuscript. Please find the detailed responses below and the corresponding corrections highlighted in red fonds in the re-submitted files.
All changes in the text consistent with the comments of reviewer 2 are in green font.
Introduction
In the introduction I suggest the authors add more details on the procedure of excision of necrosis to the fascia, tangential excision of necrosis, and split thickness skin grafting so that the reader can appreciate more what these procedures involve.
Brief explanation of surgical techniques added in lines 38-44.
In the introduction, I suggest the authors make the connection between CRP levels and sepsis clearer. For example, include, if possible, more references that provide evidence that CRP levels can be used to predict sepsis.
I have included literature regarding the use of CRP level as a prognostic marker for sepsis ( 82-84)
Pierrakos C, Velissaris D, Bisdorff M, Marshall JC, Vincent JL. Biomarkers of sepsis: time for a reappraisal. Crit Care. 2020 Jun 5;24(1):287. doi: 10.1186/s13054-020-02993-5. PMID: 32503670; PMCID: PMC7273821.
Johnson K, Messier S. Early Onset Sepsis. S D Med. 2016 Jan;69(1):29-33. PMID: 26882580.
Huang YH, Chen CJ, Shao SC, Li CH, Hsiao CH, Niu KY, Yen CC. Comparison of the Diagnostic Accuracies of Monocyte Distribution Width, Procalcitonin, and C-Reactive Protein for Sepsis: A Systematic Review and Meta-Analysis. Crit Care Med. 2023 May 1;51(5):e106-e114. doi: 10.1097/CCM.0000000000005820. Epub 2023 Mar 6. PMID: 36877030; PMCID: PMC10090344.
An X, Zhang X, ShangGuan Y. Application of PCT, IL-6, CRP, and WBC for Diagnosing Neonatal Sepsis. Clin Lab. 2023 Aug 1;69(8). doi: 10.7754/Clin.Lab.2022.220737. PMID: 37560860.
Materials and Methods
The materials and methods do not state how many patients were in each study group, when the blood samples were collected and from which body site.
Blood was collected from the unburned part of the body through a separate intravenous line or from a central venous catheter. (379-381)
I have added the information about the numer of patients in each group and study group characteristic.
The methods section could contain further information about informed consent.
Information about "patient informed consent" added in lines (378-379)
Suggest expanding the study to include more participants. Cytokine levels or other parameters could also be measured.
The results of these preliminary studies encouraged us to continue and include new parameters, which we will aim to demonstrate in a future publication. An analogous explanation was included in the text (330-334).
Results
Suggest presenting the results as charts showing mean CRP levels +/- standard deviation.
Numbers should be written with a decimal point rather than a comma.
The values that are statistically significant could be more clearly marked in the results section.
All changes have been made in the „Results” chapter
The conclusions seem to be supported by the results and references are relevant.
Round 2
Reviewer 1 Report
Comments and Suggestions for Authors
After thoroughly reviewing the revised manuscript and considering the authors' revisions and responses to the referee's comments, I find that the manuscript has been significantly improved. The authors have effectively addressed the concerns, enhancing their study's clarity and scientific rigor. The revisions have clarified the methodology, improved the presentation of results, and strengthened the discussion and conclusions.
Therefore, I believe that the manuscript now meets the standards required for publication in IJMS and I recommend that it be accepted for publication.
Thank you for considering my recommendation.
Author Response
Thank you very much for the positive review.
Kind regards.
Anna Chrapusta
Reviewer 2 Report
Comments and Suggestions for Authors
Improvements have been made to the article and suggested information has been included.
The inclusion of charts is helpful, however, figures 3 and 4 are overlapping.
Figure legends should be positioned in the most appropriate location.
Author Response
Thank you very much for your review and your time. I have improved the layout of the tables in the text and made additional changes to the blue font.
